# Establishment of Continuous *In Vitro* Culture of *Babesia gibsoni* by Using VP-SFM Medium with Low-Concentration Serum

Dongfang Li,[a,b] Lingna Wang,[a,b] Xingai Guan,[a,b] Sen Wang,[a,b] Qin Liu,[a,b] Fangwei Chen,[a,b] Yaxin Zheng,[a,b] Lan He,[a,b] Junlong Zhao[a,b]

[a]State Key Laboratory of Agricultural Microbiology, College of Veterinary Medicine, Huazhong Agricultural University, Wuhan, China
[b]Key Laboratory of Preventive Veterinary Medicine in Hubei Province, Wuhan, China

Dongfang Li and Lingna Wang contributed equally to this work. Author order was determined by project-participation time in sequential order.

**ABSTRACT** The establishment of *in vitro* culture methods has greatly facilitated the research of *Babesia*. However, the current *Babesia gibsoni in vitro* culture medium requires high concentrations of canine serum, which intensively limits the culture and is unable to satisfy the demands of long-term studies. In this study, AlbuMAX I (2 mg/mL) and 2.5% dog serum (vol/vol) were added to VP-SFM medium to develop a low-concentration serum culture medium named VP-SFMAD (2.5%), and the effectiveness of this medium was assessed by the growth of *B. gibsoni*. The results showed that VP-SFMAD (2.5%) could support the continuous growth of the parasite, and the parasitemia has no difference with the cultivation in RPMI 1640 with 20% dog serum. In contrast, either a low concentration of dog serum or absence of AlbuMAX I will significantly lower the parasite growth or fail to maintain *B. gibsoni* growth in the long term. The strategy of reducing the hematocrit was also evaluated, and VP-SFMAD (2.5%) improved the parasitemia to over 50% within 5 days. The high parasitemia is helpful for larger numbers of parasite collection, which is valuable for studying the biology, pathogenesis, and virulence of *Babesia* and other intraerythrocytic parasites. In addition, VP-SFMAD (2.5%) medium was successfully used for monoclonal parasite screening, which obtained monoclonal strains with parasitized erythrocytes about 3%, which is similar to RPMI-1640D (20%) medium that obtains monoclonal strains on the 18th day. Those results showed that VP-SFMAD can be applied to *B. gibsoni* continuous long-term, expansion culture, and subclone culture.

**IMPORTANCE** The VP-SFM as a base medium supplemented with AlbuMAX I and a low concentration of canine serum (2.5%) allowed the continuous *in vitro* culture of *Babesia gibsoni* at both small and large volumes, which was to meet different experimental needs, such as long-term culture and obtaining high parasitemia and subclone culture. The establishment of *in vitro* culture systems allows researchers to better understand the metabolism and growth patterns of *Babesia*. Importantly, several technical problems impeding such studies have been overcome.

**KEYWORDS** *Babesia gibsoni*, *in vitro*, long-term culture, VP-SFM, low serum

Babesiosis is a tick-borne haemo protozoan disease affecting a wide variety of wild and domestic animals, including cattle, horses, dogs, and rodents (1). During tick feeding, sporozoites are transmitted through saliva secretion to the vertebrate and subsequently into the bloodstream, invade erythrocytes, undergo repeated replication that differentiates into variety of morphology, and then escape and invade new erythrocytes, destroying host red blood cells (2). Continuous *in vitro* culture systems can be used to study the proliferation stage of apicomplexan parasites; these systems allow for the investigation of various aspects of parasite biology and provide useful tools for screening potential drug candidates. The establishment and development of *in vitro*

Address correspondence to Junlong Zhao, zhaojunlong@mail.hzau.edu.cn, or Lan He, helan@mail.hzau.edu.cn.

The authors declare no conflict of interest.

culture greatly promoted the establishment and application of positive genetics of the *Babesia* species.

*In vitro* cultures of *Babesia bovis* have been reported since 1980 (3). Researchers have been able to clone strains of *B. bovis* in the investigation of diagnostics, immunity, and treatment related to babesiosis owing to the continual *in vitro* growth of this parasite (4). Several benefits of culturing *in vitro* have been introduced to this protozoan parasite, but optimal nutritional requirements are not yet fully recognized. Various studies have described media with the addition of insulin, transferrin, albumin, hypoxanthine, or lipid mixtures (5–7).

The establishment of a continuous *in vitro* culture in red blood cells is a crucial step toward a deeper understanding of the fundamental biological mechanisms that govern the development of the *Babesia* parasite. Previous attempts to culture *Babesia gibsoni in vitro* culture have been successful, and the medium contains essential components for *B. gibsoni* proliferation, including nutrients, growth factors, hormones, pH regulation, and osmotic pressure (8–10). The maintenance of *in vitro* cultures of *Babesia* is still limited by a variety of factors, including medium, serum, and red blood cells. However, such work requires permanent husbandry of canines and periodic collections of blood, causing unnecessary distress to the animals and a sizable amount of human and material resources. *B. gibsoni*, in particular, requires supplement facts derived from the serological configuration of dogs. Here, new difficulties are introduced, including the increasing price of experimental dogs, food and feeding environment costs, and animal welfare.

Serum-free cultures for recent applications to parasites and *Babesia* appear to be a good option. Previous modifications to the culture medium of *Babesia in vitro* were made by reducing or removing the serum component and replacing it with other components, such as a mixture of lipids or bovine serum albumin rich in lipids called AlbuMAX I and II (7, 11, 12). The VP-SFM medium allowed the continuous *in vitro* proliferation of *Toxoplasma gondii*, *B. bovis*, and *Babesia bigemina* (13–15). Nevertheless, VP-SFM and AlbuMAX have never been evaluated for the *in vitro* culture of *B. gibsoni*. The development of new methods will go some way toward resolving the issues we now face during *in vitro* culture. Here, we report the new program of *in vitro* culture of *B. gibsoni* in dog red blood cells and evaluated its feasibility in different experimental needs, such as long-term culture and subcloned culture.

## RESULTS

**Culture initiation and continuous growth of *B. gibsoni*.** *B. gibsoni* WH strain was initially successfully cultured in RPMI 1640 medium with 40% dog serum (DS) in 10% hematocrit and incubated at 37°C under a 5% $CO_2$ incubator. In long-term culture, RPMI 1640 medium with 40% dog serum was used for 0 to 250 days. Serum concentration was reduced to 20% in 251 to 380 days. The results showed that 20% DS could also maintain the percentage of parasitized erythrocytes (PPE) between 5 and 10%, which is similar to the PPE of cultured 40% DS (Fig. 1). The growth rate of parasites were 2% ($\pm$0.6) and 2.4% ($\pm$0.6) in RPMI-1640D (40%) and RPMI-1640D (20%) medium, respectively. Importantly, the average growth rate using RPMI-1640D (20%) is fast than RPMI-1640D (40%) ($P = 0.0175$), which suggests that the parasite has adapted to RPMI-1640D (20%). According to the growth rate, the doubling time of *B. gibsoni* was roughly 6.4 to 13.7 h in the asexual stage. The parasite morphologies were observed and counted in RPMI 1640 medium with 20% dog serum. The shapes were mainly in ring; with the increase of PPE, the proportion of dyad and tetrads increased, and even a very small number of octoploids and schizonts were observed on the third day of subculture.

**VP-SFMAD supports the growth of *B. gibsoni*.** Current culture techniques require high concentrations of canine serum, which significantly raises our expenditures and restricts the amount of the *B. gibsoni* culture. Low-serum or even serum-free cultures can not only solve the problems but also reduce the difficulty in studying parasite biology and parasite-host interactions. In order to develop a low-concentration culture medium, different concentrations of dog serum were added into VP-SFM (Table 1). Parasite growth was

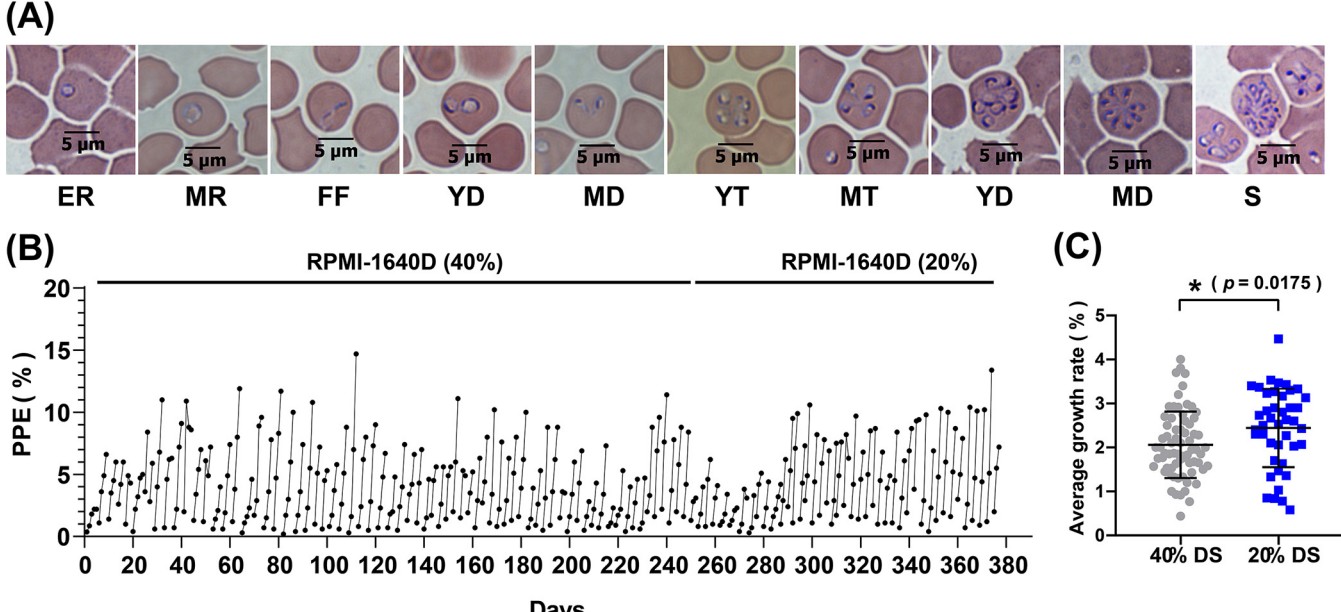

**FIG 1** *In vitro* culture initiation and continuous of *Babesia gibsoni* using RPMI 1640 medium with 40% or 20% dog serum (DS). (A) Different morphologies of *B. gibsoni in vitro* culture. (B) The parasitemia of *B. gibsoni* per subculture over 380 days (40% dog serum for 0 to 250 days and 20% dog serum for 251 to 380 days). Fresh red blood cells (RBCs) were added to decrease the parasitemia when the parasite grows for 3 or 4 days. (C) The average growth rate of each subculture using RPMI-1640D (40%) and RPMI-1640D (40%) medium. The average growth rate of every day in each subculture is obtained by the percentage of parasitized erythrocytes (PPE) of last day subtracting the PPE after been diluted involved in the previous subculture and then dividing the number of days cultivated in each subculture. ER, early rings; MR, mature rings; FF, filamentous forms; YD, young dyad; MD, mature dyad; YT, young tetrads; MT, mature tetrads; YO, young octoploid; MO, mature octoploid; S, schizont.

assessed by comparing PPE for a total of 15 days (Table 1; Fig. 2A). The results showed that VP-SFMA could support the growth of *B. gibsoni*, but the PPE was significantly lower than RPMI-1640D (20%) ($P < 0.0001$), reaching only around 3% on the third day (Fig. 2B). With the addition of 2.5%, 5%, and 20% dog serum in VP-SFM medium together with 2 mg/mL of AlbuMAX I, the maximum PPE could reach 6.0%, 6.7%, and 6.5% on day 3, respectively, and the growth rates of *B. gibsoni* in the three groups were similar to that of the group with RPMI-1640D (20%) ($P > 0.05$). However, the PPE in the group with VP-SFMAD (1.25%) could not reach 5% on the third day, indicating that 1.25% of dog serum is too low to support the growth of *B. gibsoni*. The growth of *B. gibsoni* could not be maintained with RPMI-1640D (2.5%) or RPMI-1640F (20%), with the PPE decreasing, eventually to less than 0.1%. Continuous growth was possible with RPMI-1640AD (2.5%), but the PPE reached no more than 3% by the third day, which was slower than when only 20% dog serum was added to the medium (Fig. 2B). However, when using VP-SFMAD (2.5%) medium for *in vitro* culture of *B. gibsoni*, instability in multiplication rates was observed. It is possible that the parasite requires a long period of adaptation, similar to the previous

**TABLE 1** Ingredients of medium used in *in vitro* culture of *B. gibsoni*[a]

| Group | Ingredients | | |
| --- | --- | --- | --- |
| | Medium | AlbuMAX I | Serum |
| VP-SFMA | VP-SFM | 2 mg/mL | 0 |
| VP-SFMAD (1.25%) | VP-SFM | 2 mg/mL | 1.25% DS |
| VP-SFMAD (2.5%) | VP-SFM | 2 mg/mL | 2.5% DS |
| VP-SFMAD (5%) | VP-SFM | 2 mg/mL | 5% DS |
| VP-SFMAD (20%) | VP-SFM | 2 mg/mL | 20% DS |
| RPMI-1640AD (2.5%) | RPMI-1640 | 2 mg/mL | 2.5% DS |
| RPMI-1640D (2.5%) | RPMI-1640 | 0 | 20% DS |
| RPMI-1640D (20%) | RPMI-1640 | 0 | 20% DS |
| RPMI-1640F (20%) | RPMI-1640 | 0 | 20% FBS |

[a]DS, dog serum; FBS, fetal bovine serum.

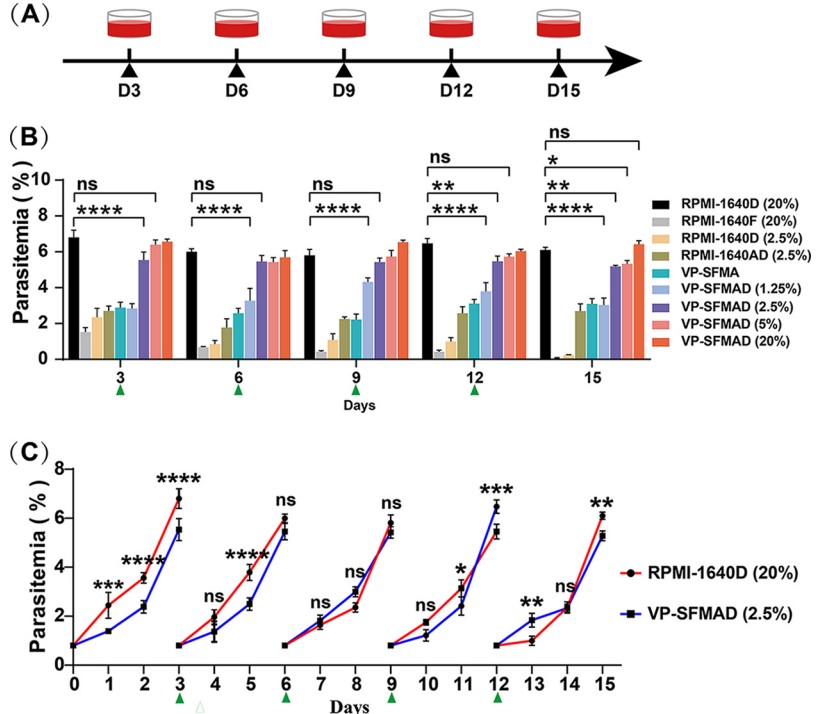

**FIG 2** Gradual lowering of the serum concentration and AlbuMAX I in the culture medium for its elimination from the *in vitro* culture of *B. gibsoni*. (A) Pattern diagram of five subcultures of *in vitro* culture. (B) The serum concentration was lowered from 20% to 0% in VP-SFM medium with 2.5 mg/mL AlbuMAX I. RPMI-1640D (20%) was the control, and RPMI-1640F (20%) and RPMI-1640D (2.5%) were analyzed at the same time. The data represent mean values measured in triplicate. (C) Changes in parasitemia for 15 days in in vitro culture using RPMI-1640D (20%) and VP-SFMAD (2.5%). ns, no significant difference; *, $P \leq 0.05$; **, $P < 0.01$; ***, $P < 0.001$ (significant difference); ****, $P < 0.0001$ in comparison with RPMI-1640D (20%) group.

reduction of the serum concentration from 40% to 20%. Therefore, we subsequently used VP-SFMAD (2.5%) medium for long-term culture. In conclusion, poor growth of *B. gibsoni* could be maintained by VP-SFM medium supplemented with AlbuMAX I, and the maximal growth rates of *B. gibsoni* could be restored with 2.5% dog serum supplementation.

**VP-SFMAD (2.5%) medium supports the long-term culture of *B. gibsoni*.** *B. gibsoni* was cultured in VP-SFMAD (2.5%) for 60 days to determine whether the medium supported the continuous growth of *B. gibsoni*. RPMI-1640D (20%) medium was used as a control (Fig. 3A). The results showed that VP-SFMAD (2.5%) was able to maintain the growth of *B. gibsoni* with a maximum PPE of 8.5%, and there was no statistically significant difference ($P > 0.05$) from the group with RPMI-1640D (20%) (Fig. 3B). The parasites collected from both media showed normal morphology on microscopic analysis (Fig. 3C).

**VP-SFMAD (2.5%) could support expansion for high parasitemia.** When culturing *B. gibsoni* in 10% hematocrit, the maximum PPE is usually about 8%; then the growth of the parasite started to deteriorate, and replication slowed down. However, high parasitemia is needed sometimes for biological studies such as obtaining high-quality parasite antigens and collecting a large number of parasites in transfection and omics studies. In this study, VP-SFMAD (2.5%) was subjected to high parasitemia culture. The results showed that VP-SFMAD (2.5%) could be used for *B. gibsoni* expansion and supported high parasitemia. There is no significant difference with the parasitemia by using RPMI-1640D (20%) (Fig. 4). When the PPE reached about 8%, 1,000 $\mu$L medium was added to the well, and one well was distributed into two wells. On the fifth day, the PPE reached 20%, 1,000 $\mu$L medium and 20 $\mu$L fresh red blood cells (RBCs) were added to each well, and two wells were split into four wells. The parasitemia could reach more than 50% on the seventh day (Fig. 4B). Importantly, on the fifth day, the proportion of ring parasites decreased; the proportion of dyads, tetrads, and octoploids increased significantly

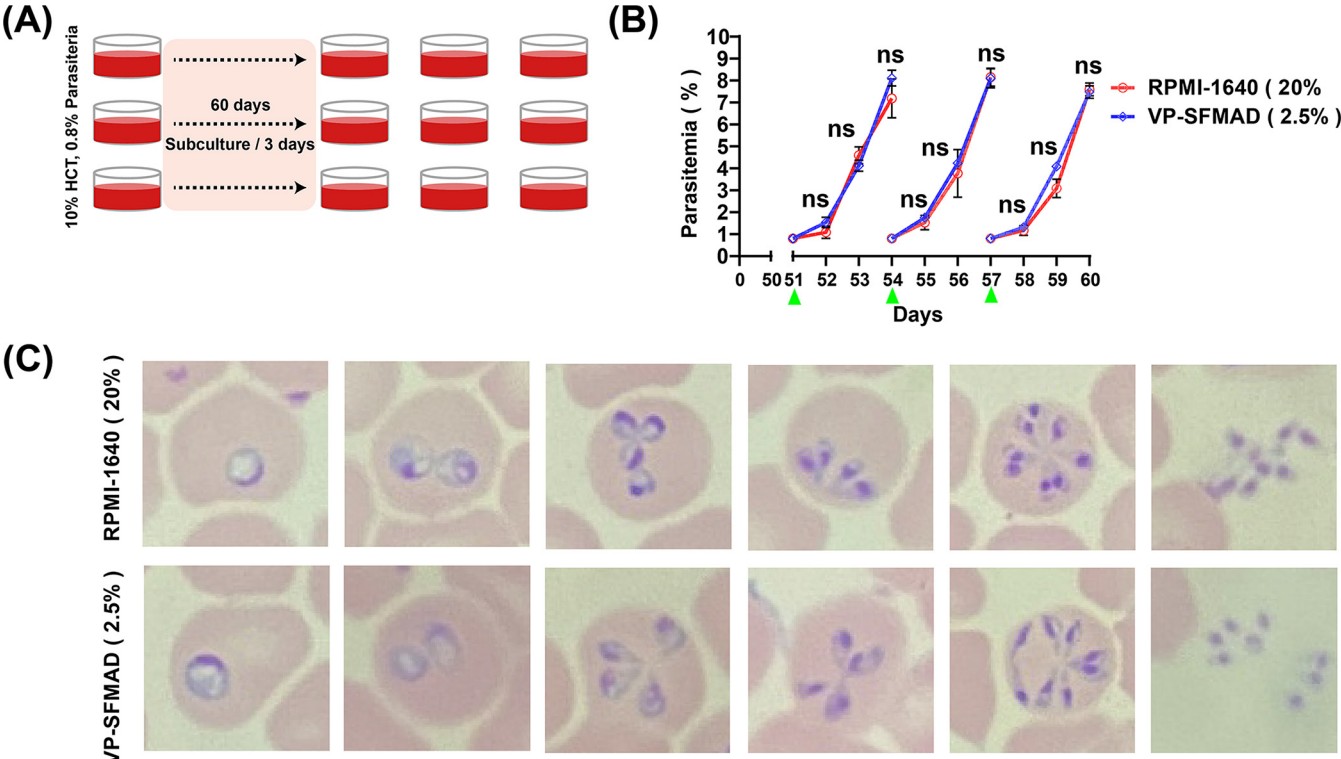

**FIG 3** Long-term culture of *B. gibsoni* using VP-SFMAD (2.5%) and RPMI-1640D (20%). (A) Pattern diagram of long-term culture. (B) The parasitemia of *B. gibsoni* in the last three subculture of 60 days of long-term culture by counting of Giemsa-stained blood smears, and 4,000 RBCs were counted. The data represent mean values measured in triplicate. ns, no significant difference; *, $P \leq 0.05$; **, $P < 0.01$; ***, $P < 0.001$ (significant difference); ****, $P < 0.0001$ in comparison with RPMI-1640D (20%) group. (C) Giemsa-stained blood smear showed morphologies of *B. gibsoni* using VP-SFMAD (2.5%) and RPMI-1640D (20%).

(Fig. 4C); and no morphological alterations were observed during microscopic analysis for both cultures.

**Derivation of a subclone of ΔBgef1a-b isolate using VP-SFMAD medium.** Subclonal screening is crucial and commonly used after transfection to investigate the phenotypic of genetically changed strains. VP-SFMAD (2.5%) was used for subclonal screening of Bgef1a-b knockout strains to evaluate whether the new medium could support a single parasite or low parasitemia propagate. The plasmid (pBS-ef1a-B) expressing mCherry and PAC was constructed, using 750-bp 5′-untranslated region (UTR) and 3′-UTR of ef-1α B as the promoter and the terminator, respectively (Fig. 5A). Bgef1a-B 5′-UTR and Bgef1a-B 3′-UTR were used as recombination sites cloned into the upstream and downstream of mCherry-PAC gene to construct plasmid pBS-ef1a-B. At 12 days postselection by 4 nM puromycin, mCherry-expressing parasites appeared in cultures transfected with plasmids (Fig. 5B). PCR1 (F1R1), PCR2 (F1R2), and PCR3 (F2R1) primer pairs could successfully amplify 3,346-, 2,329-, and 2,322-bp fragments in the ΔBgef1a-B strain, respectively, but not in the wild-type (WT) strain of *B. gibsoni* (Fig. 5C). Western blot analyses indicated the expression of protein mCherry-PAC in Bgef1a-b knockout strains (Fig. 5D). After 24 days selected. the culture was serially diluted to 30 parasites in 6 mL volume and subsequently aliquoted to 60 wells in a 96-well plate, which suggested 0.5 parasite/well. The medium was changed every 3 days until 18 days, and the PPE reached about 3% (Fig. 6A). The cultures were transferred to a 24-well plate, respectively. The clones were able to reach a PPE of 8% after 3 days in VP-SFMAD (2.5%), which was consistent with our usual protocol for obtaining subclones with RPMI-1640D (20%). In addition, red fluorescence was detected in almost all parasites using flow cytometry, which indicated that parasites obtained using VP-SFM AD (2.5%) medium were monoclonal after 18 days of culture without drug screening (Fig. 6B).

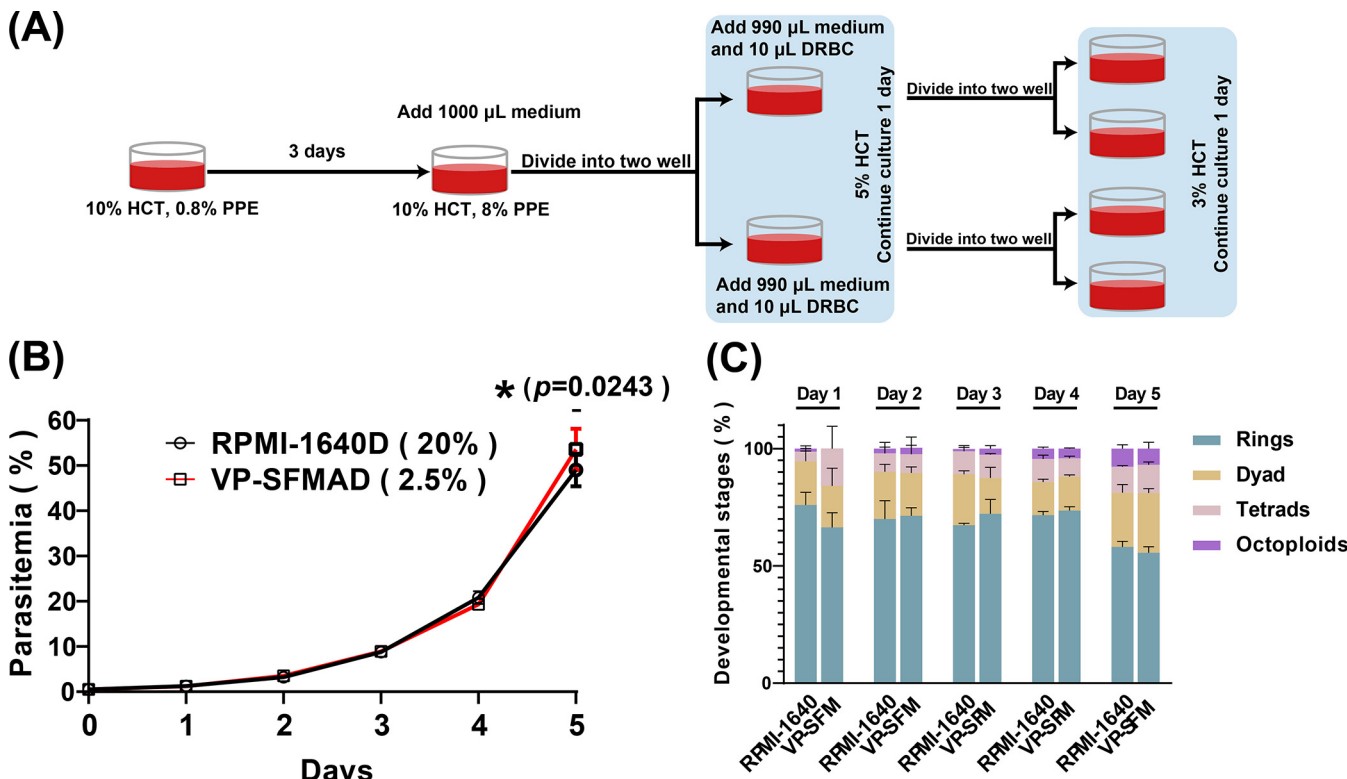

**FIG 4** *B. gibsoni* expansion by using VP-SFMAD (2.5%) and RPMI-1640D (20%) medium. (A) Pattern diagram of protocols to increase PPE by lowering hematocrit. (B) Parasitemia of *B. gibsoni in vitro* culture in 5 days using RPMI 1640 and VP-SFM, and hematocrit (HCT) was lowered on the third and fourth days. The data represent mean values measured in triplicate. ns, no significant difference; *, $P \leq 0.05$; **, $P < 0.01$; ***, $P < 0.001$ (significant difference); ****, $P < 0.0001$ in comparison with RPMI-1640D (20%) group. (C) Percentages of different parasite development stages as identified in VP-SFMAD (2.5%) and RPMI-1640D (20%). The data are presented as mean standard deviation (SD) of three independent experiments performed in three biological replicates. No significant differences were observed between the different developmental stages in the two groups ($P < 0.001$).

## DISCUSSION

Due of the lack of a trustworthy culture method, it is challenging to investigate the sexual stage of *Babesia* in the tick; however, red blood cell culture can be used to keep the asexual stage of *Babesia* alive. Here, we report the initiation and continuous culture of *B. gibsoni* in RPMI 1640 medium with a high proportion of dog serum (40% and 20%) in dog red blood cells. The establishment of *Plasmodium* cultures provides reference data for *in vitro* culture of *Babesia* species, during which several cultural factors are identified as key factors: For instance, adequate aseptic technique management, leukocyte removal from cells, appropriate culture temperature, glucose addition, anaerobic conditions, and liquid level (3). For the investigation of *Babesia* culture, the donor cell is mature erythrocytes that provide the appropriate environment and nutrition requirements for intraerythrocytic-stage growth and replication. Red blood cells from carriers of the same species are considered a determining factor, as is the proliferation of *Babesia microti*, and *B. microti* can only be obtained in hamster erythrocytes if the medium is supplemented with bovine serum, rat serum, or hamster serum (16).

Regarding the initial continuous culture of *B. bovis*, a stirred suspension culture device with daily medium changes and a culture method consisting of a fixed layer of erythrocytes are used, known as microaerophilic stationary-phase culture technology (3, 17, 18). Further improvement of the isolation treatment method of red blood cells and adjustment of culture conditions according to pH value, buffer composition, and medium exchange frequency, to develop optimized conditions for continuous culture of *B. bovis* (19). The culture of *B. bovis* is further optimized by using chemically defined medium that is independent of bovine serum (6). Therefore, *B. bovis* represents the most distinctive *Babesia* culture program that in now available. Other *Babesia* culture procedures, such as *Babesia divergens*, have found that it can be grown in a serum-free

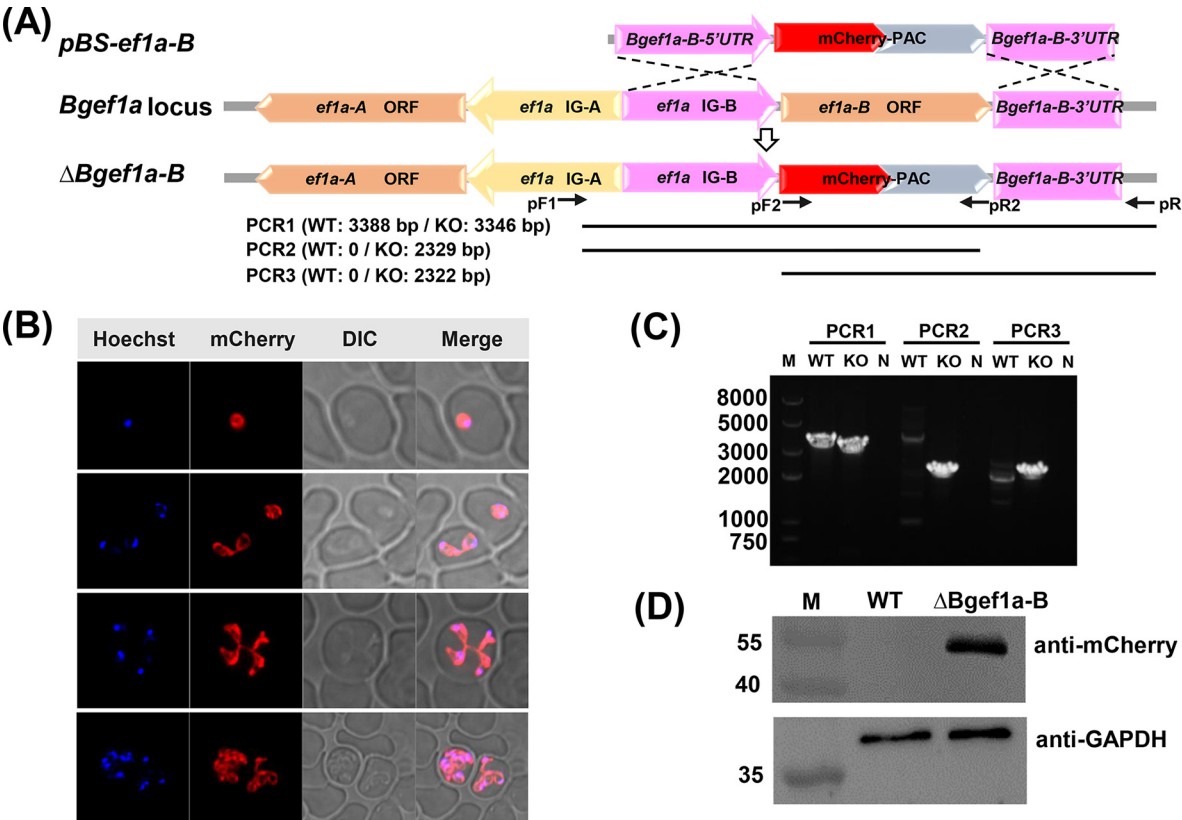

**FIG 5** Establishment of the Bgef1a-b knockout strains with stable expression of mCherry-PAC. (A) Pattern diagram of subcloned parasite. (A) Plasmid construct of pBS-ef1a-B showing the recombination sites for integration into ef-1α locus by double crossover homologous recombination. (B) Hoechst staining represents the nucleus of the parasite, red fluorescence corresponds to the transfected parasite expressing mcherry, and the DIC image shows a parasitized RBC. The merged image represents the overlap of all images. (C) PCR confirmation of the integration of pBS-ef1a-B into the ef-1α locus. ΔBgef1a-B strains were identified by PCR1, PCR2, and PCR3, with the wild-type (WT) strain used as the control. (D) Western blot analyses of Bgef1a-b knockout (KO) strains. The express of mCherry-PAC was identified using anti-mCherry, with anti-GAPDH used as internal reference. ORF, open reading frame; UTR, untranslated region.

medium, but parasitemia is significantly reduced (11). A culture protocol for *Babesia caballi* infected with horses was initially established using equine serum and equine erythrocytes and cultured in a serum-free medium with a parasitemia of 4% to 6% (7). Another species of horse-infected *Babesia* was also cultured *in vitro*, using a special concentrated medium containing additives such as insulin, transferrin, and saturated and unsaturated fatty acids (5). The same attempt to use serum-free cultures resulted in a reduction of only 2% to 4% of infected erythrocyte parasitemia (7). These results suggest that serum is not necessary for *in vitro* culture of *Babesia* in appropriate culture systems but can increase parasitemia. Therefore, it is possible to establish a low or even serum-free culture of *B. gibsoni*.

Although the RPMI 1640 medium is an inexpensive and easily accessible basal medium, our access to dog serum is restricted due to the high expense of buying and breeding laboratory dogs. More importantly, not all dog sera can be utilized for the culture of *B. gibsoni*. In addition, dog serum shortages worsened during the COVID-19 pandemic with further increases in the cost of dog breeding. Faced with these challenges, we used the recently widely reported medium VP-SFM, which has been used for serum-free culture of a variety of species of *Babesia*, such as *B. bovis* and *B. divergens*. We determined that *B. gibsoni* proliferated in VP-SFMAD (2.5%) medium. In contrast, the other medium formulations failed to maintain *B. gibsoni* growth. Using strategies to diminish hematocrit, this medium system could improve the parasitemia to over 50% of parasitized erythrocytes. These larger numbers of parasites harvested from *in vitro* culture can serve as an important source of biological material for the development of strategies to control *B. gibsoni*. When the serum is

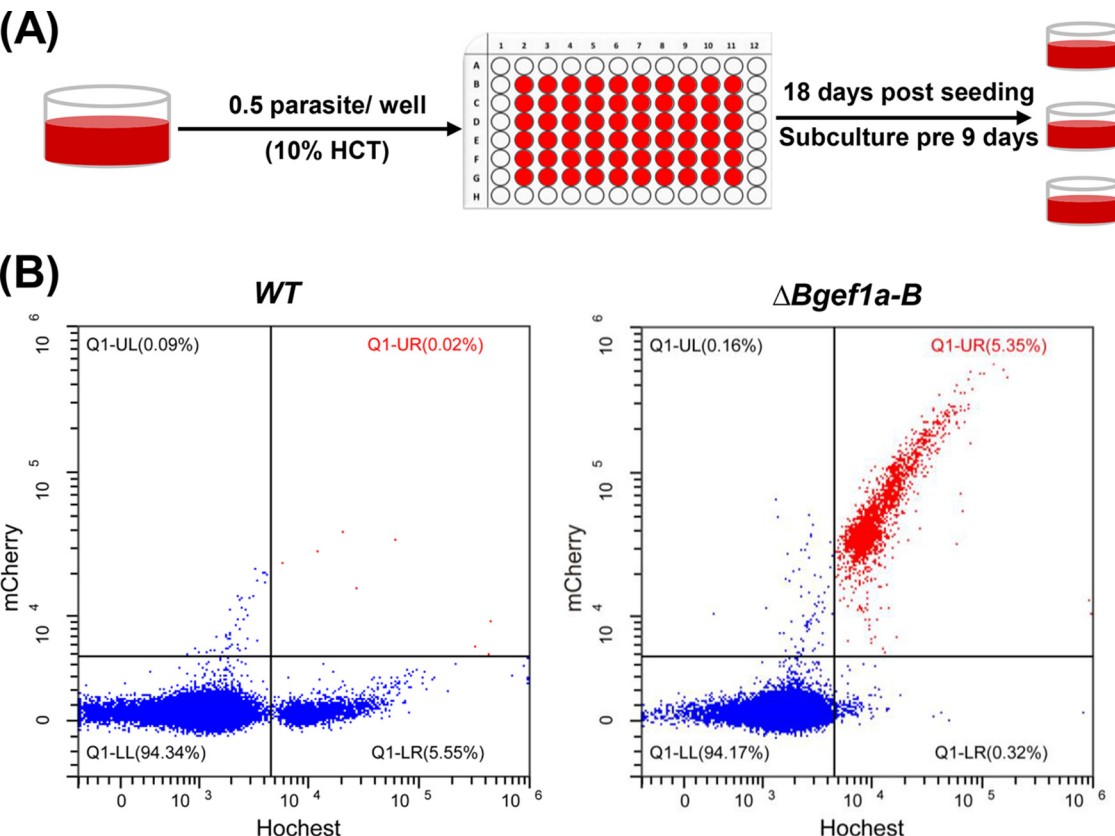

**FIG 6** Limiting dilution cloning of parasites using VP-SFMAD (2.5%). (A) Pattern diagram of subcloned parasite. (B) Flow cytometry results of WT and ΔBgef1a-B are displayed, with the abscissa for Hoechst staining and the ordinate for mCherry fluorescence.

removed, the growth of the parasite using VP-SFM medium is still observed, but at a slower PPE, indicating that the components in the serum contribute to the parasite's survival. Recently, the serum-free culture of *B. bovis* was accomplished with success using A-DMEM/F12 medium alone with the addition of insulin, transferrin, and selenite, which led to a consistent increase in PPE to 9.7% (20). Genomic analysis of *B. bovis* showed that it could not synthesize polyamines, so the parasite may obtain these compounds in the external environment of the host and add putrescine at a concentration of 0.101 mg/liter to A-DMEM/F12 medium with a maximum PPE value of 6.23% (21). Putrescine dosage needs to be very precise; concentrations below 0.05 mg/liter do not stimulate the proliferation of *Babesia*, while levels above 0.405 mg/liter seem hazardous (22). The study of nutrients and growth factors required for *in vitro* culture of *Babesia* will help us establish a more appropriate culture protocol.

In general, donor erythrocytes are suspended in a culture medium and supplemented with host serum during the *in vitro* culture of *Babesia*. Complete media are a source of nutrition for the growth and reproduction of the parasite, and through *in vitro* culture of the *Babesia* parasite, researchers can determine certain nutritional needs of this parasite (6); study different aspects of parasite biology, its life cycle (23, 24), and host-parasite relationships (25); and identify relevant proteins during the invasion. It also helps to gain knowledge about the metabolic, physiological, and reproductive behavior of this native parasite. Additionally, in immunoprophylaxis investigations for the generation of antigens for serological diagnosis and for drug susceptibility research, culture systems have been exploited as a source of soluble antigens (26). Most crucially, the use of live immunogen from *in vitro* culture has been stressed as an option in the control of babesiosis.

**TABLE 2** List of primers used in this study[a]

| Primer | Sequence (5′ to 3′) |
|---|---|
| Bgef1a5H-F | **GAATTCCTGCAGCCCGGG**CTGGAACCATCTCCTACCAAC |
| Bgef1a5H-R | **CTCGCCCTTGCTCACCAT**TTTGGTAAAGGTTGACGATAGC |
| pBs-VcF | **GTGGTTGGGATTCTCCTG**GGATCCACTAGTTCTAGAGCG |
| pBs-VcR | **GTTGGTAGGAGATGGTTCCAG**CCCGGGCTGCAGGAATTC |
| mCh-PAC-F | **GCTATCGTCAACCTTTACCAAA**ATGGTGAGCAAGGGCGAG |
| mCh-PAC-F | **CTTTAGTTAACACGAAATAATCAGCT**TTAGGCACCGGGCTTGC |
| Bgef1a3H-F | **GCAAGCCCGGTGCCTAA**AGCTGATTATTTCGTGTTAACTAAAG |
| Bgef1a3H-R | **CGCTCTAGAACTAGTGGATCC**CAGGAGAATCCCAACCAC |
| pF1 | GTAATCTACGGCAACACATAGC |
| pR1 | CTTGTCGTAGCTCAAGTGAC |
| pF2 | ATGGTGAGCAAGGGCGAG |
| pR2 | TTAGGCACCGGGCTTGC |

[a]The bold type represents the homologous arm used to construct the plasmid by homologous recombinant plasmid.

## MATERIALS AND METHODS

**Parasite.** The *B. gibsoni* WH strain was originally isolated from a naturally infected dog in Wuhan, China, in 2017. The strain underwent a 280-day domestication process before stable *in vitro* culture, during which the growth of the parasite was unstable with the highest parasitemia changed greatly from 3% to 8%, and the subculture cycle was 5 to 8 days. The isolate was stored in liquid nitrogen in the Department of Veterinary Medicine, Huazhong Agriculture University, Wuhan, China.

**Animals.** *Babesia*-free female beagle dogs were used as blood donors. Nine dogs were purchased from Hubei Yizhicheng Biotechnology Co., Ltd., and fed at the Laboratory Animal Centre, Huazhong Agriculture University. This study was approved by the Scientific Ethics Committee of Huazhong Agricultural University (permit no. HZAUDO-2019-003). All dogs were handled in accordance with the Animal Ethics Procedures and Guidelines of the People's Republic of China. The animal was kept in tick-free folds and protected from contamination with other pathogens. The animals had unlimited access to water and fodder.

**Erythrocytes and dog serum.** Blood was collected in 10-mL sterile $K_3$EDTA vacutainers from beagles. The blood was centrifuged at $650 \times g$ for 5 min at 4°C. Erythrocytes were washed three times with 10 mL RPMI 1640 medium and centrifuged as above to remove the white blood cells. The final packed RBCs were suspended in 2 volumes of RPMI 1640 and stored at 4°C until use (no more than 7 days). Normal dog serum (DS) was collected from those beagles and stored at −20°C until use.

**Culture media.** Parasite growth was monitored and evaluated in the following media: RPMI 1640 (GIBCO, Grand Island, NE, USA) and VP-SFM (GIBCO, Grand Island, NE, USA). The detailed formulas of the media are as follows: RPMI 1640 culture medium: RPMI 1640 ($1\times$), 0.2 g/liter pyruvic acid (Aladdin), 2 g/liter $NaHCO_3$ (Sigma), 2 mM L-glutamine (GIBCO), and 100 IU/mL penicillin (Corning); and VP-SFM culture medium: VP-SFM (20 g/liter), 4 mM L-glutamine (GIBCO), and 100 UI/mL penicillin (Corning). The two media were supplemented with different ratios of serum (dog serum or fetal bovine serum [FBS]) or AlbuMAX I (GIBCO) as shown in Table 1. The media were adjusted to pH 6.8 and filter sterilized through a 0.22-$\mu$m filter.

**Culture initiation and continuous.** *B. gibsoni* was cultured *in vitro* according to the protocol of Zweygarth and Lopez-Rebollar (9). Briefly, the *in vitro* culture of *B. gibsoni* started with a percentage of parasitized erythrocytes (PPE) of 1% and was maintained at 10% (vol/vol) fresh dog erythrocytes in RPMI 1640 medium consisting of 40% dog serum. The depth of media in the flasks was 7 mm, and the 37°C incubator had an atmosphere of 92% $N_2$, 5% $CO_2$, and 3% $O_2$. Approximately 800 $\mu$L of culture media overlaying the erythrocytes were removed daily and replaced with fresh medium. Subculturing was performed when any culture exceeded a PPE of about 8% and adjusted to 0.8% by the addition of uninfected erythrocytes. The serum concentration is gradually decreased via a gradient to 20% once the growth of *B. gibsoni* is stable (about 250 days), at which concentration its stable culture is maintained without altering the other culture parameters described above.

**Plasmid constructs and transfection of parasites.** The schematic diagram of the plasmid (pBS-ef1a-B) used in this study is shown in Fig. 5A, and the sequence of plasmid pBS-ef1a-B has been uploaded in National Center for Biotechnology Information (NCBI) database with the accession number OQ630424. The Bgef1a-B gene (BgWH_03g00751) located on chromosome 3 of *B. gibsoni* has been deposited with the accession number OQ630425 in NCBI. The reporter gene (mCherry) and drug selection gene (puromycin) cassettes were driven with Bgef1$\alpha$ 5′-UTR (IG-B). Bgef1$\alpha$ 5′-UTR (IG-B) and Bgef1$\alpha$ 3′-UTR were used as recombination sites cloned into the upstream and downstream of the mCherry-PAC genes. All the PCR primer pairs used for plasmid construction are listed in Table 2. The constructed plasmid was purified using Qiagen plasmid maxi kit (Qiagen, Hilden, Germany) according to the manufacturer's instructions and was confirmed by sequencing before transfection. Transfection of parasites as previously described (27). In brief, *B. gibsoni*-infected red blood cells (iRBCs) were centrifuged at low speed, and then the culture supernatant was removed. Linearized pBS-ef1a-B plasmid (20 $\mu$g) and iRBCs were mixed and transfected using Lonza buffer SF and program FA113 of the Amaxa 4D Nucleofector device (Lonza, Cologne, Germany). The transfected mixtures were immediately transferred into a preheated culture containing 10% fresh RBCs. Puromycin (4 nM) was used to select mCherry-expressing transgenic parasites. After 3 weeks of drug selection, the parasite population was cloned in a 96-well culture plate using limiting dilution as previously described (28).

**Analysis of subclone isolate by PCR, Western blot, and flow cytometry.** Two pairs of primer (Table 2) (F1 and R1; F2 and R2) were designed to confirm the integration of mCherry-PAC into the ef-1αB locus. The position of the primer and the size of the product are shown in Fig. 5B. Total proteins extracted from parasite pellets were detected with anti-mCherry (rabbit; 1:5,000; Proteintech, Shanghai, China). The antibody of GAPDH (mouse; 1:5,000; Proteintech, Shanghai, China) was used as the internal control. The ΔBgef1a-B subclone isolate was analyzed with a flow cytometer (CytoFLEX LX) to quantify the proportion of mCherry-positive parasites, and the cell nuclei were stained with 2 $\mu$g/mL Hoechst 33342 (Sigma, Shanghai, China). The data were analyzed with CytExpert 2.4 (https://cytexpert.updatestar.com/).

**Giemsa staining and statistical analysis.** The cultured samples were spread onto microscope slides, fixed with methanol, and Giemsa-stained to monitor parasite proliferation by light microscopy under 100× oil immersion. The average PPE was determined by counting 4,000 erythrocytes for each smear. Three separate experiments were carried out utilizing triplicate wells for each test condition. *B. gibsoni*. Dunnett's test was used to compare the means of different experimental groups with the mean of a control group. Additionally, an independent *t* test was performed, and significance was defined as $P < 0.05$.

**Data availability.** All data generated from the study are presented in the report. Sources of the raw data used in the analysis have been cited.

## ACKNOWLEDGMENTS

This work was supported by grant 2022YFD1801700 from the National Key Research and Development Program of China, grants 32172879 and 31930108 from the National Natural Science Foundation of China, funds from the Top-notch Young Talent Supporting Program (L.H.), and grant 2262022DKYJ001 from the Fundamental Research Funds For the Central Universities.

J.Z. and L.H. conceived and designed this study. D.L., L.W., and X.G. conducted the laboratory work. D.L. drafted the manuscript. Q.L., F.C., and Y.Z. analyzed the data. J.Z., L.H., and S.W. critically revised the manuscript. All authors have read and approved the final manuscript.

This study was approved by the Scientific Ethics Committee of Huazhong Agricultural University under permit No. HZAUDO-2019-003. All dogs were handled in accordance with the Animal Ethics Procedures and Guidelines of the People's Republic of China.

We declare no conflict of interest.

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
