## [Reviewer comments · Microbiology Spectrum]

Microbiology Spectrum

Establishment of continuous *in vitro* culture of *Babesia gibsoni* by using VP-SFM medium with low concentration serum

Dongfang Li, Lingna Wang, Xingai Guan, Sen Wang, Qin Liu, Fangwei Chen, Yaxin Zheng, Lan He, and Junlong Zhao

Corresponding Author(s): Junlong Zhao, Huazhong Agricultural University

Review Timeline:

Submission Date:	January 16, 2023
Editorial Decision:	February 7, 2023
Revision Received:	February 25, 2023
Editorial Decision:	March 9, 2023
Revision Received:	March 16, 2023
Accepted:	April 4, 2023

Editor: Kevin Tan

Reviewer(s): Disclosure of reviewer identity is with reference to reviewer comments included in decision letter(s). The following individuals involved in review of your submission have agreed to reveal their identity: Benoit Malleret (Reviewer #2)

Transaction Report:

DOI: <https://doi.org/10.1128/spectrum.00258-23>

February 7, 2023

Prof. Junlong Zhao
Huazhong Agricultural University
College of Veterinary Medicine
Shizishan street 1
Wuhan, Hubei 430070
China

Re: Spectrum00258-23 (Establishment of continuous *in vitro* culture of *Babesia gibsoni* by using VP-SFM medium with low concentration serum)

Dear Prof. Junlong Zhao:

The manuscript has been reviewed by 2 experts in the field. While they find the work interesting in the context of Babesia culture and subsequent applications, both reviewers highlighted the need for statistical validation of the results. Reviewer #1 pointed out the need for an improvement in written English and reviewer #2 raised concerns on the lack of robust evidence for the generation/ purity of clones (the need for sequencing, flow cytometry etc). Hence, a major revision is required before a resubmission for review can be made.

Link Not Available

Sincerely,

Kevin Tan

Journals Department
Reviewer comments:

Reviewer #1 (Comments for the Author):

The study of babesial parasites is greatly enhanced by the ability to cultivate parasite asexual stages in *in vitro* culture. Prior methodology for the cultivation of *B. gibsoni* were not adequate to perform some of the manipulations needed to delve into the

biology of the parasite, a problem that the authors appear to have solved. The title and abstract reflect accurately the content of the manuscript and the figures are of acceptable quality (with the exceptions noted). Rewriting to improve English usage is required before this manuscript could be published. No new information on the biology of *B. gibsoni* is provided, but this manuscript could provide useful information on culture methodology for raising *B. gibsoni* for study. The manuscript would benefit by addressing the issues below.

1. The Abstract should be rewritten to shorten it by about 1/3. Lines 52-56 could be eliminated altogether without loss. Lines 56-58 could be rewritten as, "Importantly, several technical problems impeding such studies have been overcome."
2. Line 64. "...different numbers.." is meaningless. Please rephrase for meaning. Also, briefly mention what happens once merozoites have invaded the red cell.
3. Line 66. What is meant by, "...value-added phase of apical compound parasites.."? It is not clear what the authors were saying.
4. Lines 68-70 could be deleted without loss.
5. Line 73. Reference (2), referring to the seminal paper by Levy and Ristic, is not even included among the references.
6. Lines 80-81 could be deleted without loss.
7. Lines 94-97 could be deleted without loss.
8. Line 110. Title should read, "...continuous growth of *B. gibsoni*."
9. Why was the experiment shown in Figure 1 not performed as a direct comparison of the two media compositions? It is not possible to directly compare the growth rates of parasites at different times in different media, perhaps with different dog sera. Given how this was done, it could be possible to perform a Z-test analysis on normalized rates of proliferation in the two media, and plotted as cumulative proportions. This would at least provide statistical support for the growth rates being equivalent (or not), although from an experimental point of view it is an inferior approach.
10. Figure 1. Could the authors explain why so many parasites have a highly vacuolated appearance?
11. Lines 131, 207. It would be helpful to state the cell cycling time for this parasite as measured in vitro.
12. Lines 138-139 (and throughout). The data should be stated with calculated s.d. or s.e. values. When it is stated that, "...growth rates of *B. gibsoni* in the three group had no difference.." no measure of reproducibility or statistical support was given for this statement.
13. Figure 2. Labeling on this figure is much too small to read. Also, it is incorrect to refer to the five subcultures as "...five consecutive generations..". Over 15 days these parasites likely went through 30-50 generations.
14. Figure 3B. Figure labeling is much too small to read.
15. Lines 187-190. The statement about octoploids arising from daughter merozoites not leaving the cell may be true, but some form of actual experimental evidence is required to support that statement. The alternative, that multiple merozoites can invade the same erythrocyte, was not ruled out.
16. Figure 5. What is the source of the mCherry fluorescence? Clearly, these are genetically modified parasites, which is incompatible with the description of parasites in the Methods section. The nature of the modifications and the cultivation history of these parasites must be identified, particularly as these parasites have previously undergone significant in vitro culture adaptation. Also, there is a significant pixel-shift problem with Figure 5B where light and fluorescence images do not align properly. This must be corrected.
17. Line 224. Reference 15 does not refer to continuous culture. This was not accomplished with *Babesia* spp. until Levy and Ristic (1980) achieved this for *B. bovis*.
18. Line 233. "microaerobic.." should be "microaerophilous..", and the correct reference is again Levy and Ristic (1980).
19. Line 291. It should be stated when the parasites were isolated in Wuhan, and any prior cultivation history. This sounds as though the authors were working with a primary isolate, but this is clearly not the case. Figure 5 dispels that idea.
20. Table 1 could be eliminated, as the information is provided in the text. Alternatively, refer to Table 1 and eliminate most of the associated verbiage from the text.
21. Line 334. "emersion" should be "immersion".

Reviewer #2 (Comments for the Author):

The manuscript "Establishment of continuous in vitro culture of *Babesia gibsoni* by using VP-SFM medium with low concentration serum" by Li et al. describes the methodology for a long-term in vitro culture of a dog *Babesia* species. The article is well written, but some statistical and technical details are missing particularly for the subcloning.

Major points:

Figure 2B and 2C: The authors must add the statistical analysis.

In the result par on the derivation of subclone of Delat Bgef1a-b isolate, the authors must have full genomic data to demonstrate the success of the subclone isolation.

Figure 5B: The authors need to add flow cytometry data to show the purity of the fluorescent subclone strains.

Materials and Methods: The sections for the generation of Delta Bgef1a-b transgenic parasites by electroporation and the section for the fluorescence microscopy are missing.

Minor points:

Line 42: Define PPE, it should be for parasitized erythrocytes.

Figure 1B: The authors should highlight in the figure the days with 40% and 20% DS and in the legend specify when fresh RBC are added to decrease the parasitemia.

Figure 2B: The authors must add the number of biological replicates in the legend.

Line 308: Any references to cite showing that penicillin has no effect on Babesia parasite development?

Staff Comments:

Preparing Revision Guidelines

Please return the manuscript within 60 days; if you cannot complete the modification within this time period, please contact me. If you do not wish to modify the manuscript and prefer to submit it to another journal, please notify me of your decision immediately so that the manuscript may be formally withdrawn from consideration by Microbiology Spectrum.

Reviewer #1 (Comments for the Author):

The study of babesial parasites is greatly enhanced by the ability to cultivate parasite asexual stages in *in vitro* culture. Prior methodology for the cultivation of *B. gibsoni* were not adequate to perform some of the manipulations needed to delve into the biology of the parasite, a problem that the authors appear to have solved. The title and abstract reflect accurately the content of the manuscript and the figures are of acceptable quality (with the exceptions noted). Rewriting to improve English usage is required before this manuscript could be published. No new information on the biology of *B. gibsoni* is provided, but this manuscript could provide useful information on culture methodology for raising *B. gibsoni* for study. The manuscript would benefit by addressing the issues below.

Response: We feel great thanks for your professional review work on our article that we have used to improve the quality of our manuscript. According to your constructive suggestions, we have made extensive corrections to our previous draft, the detailed corrections are listed below.

1. The Abstract should be rewritten to shorten it by about 1/3. Lines 52-56 could be eliminated altogether without loss. Lines 56-58 could be rewritten as, "Importantly, several technical problems impeding such studies have been overcome."

Response: Thank you for the suggestion. We have removed lines 52-56 and re-written this part to shorten it according to the Reviewer's suggestion, and Lines 56-58 has been rewritten as "Importantly, several technical problems impeding such studies have been overcome."

2. Line 64. "...different numbers.." is meaningless. Please rephrase for meaning. Also, briefly mention what happens once merozoites have invaded the red cell.

Response: According to the Reviewer's suggestion We have re-written this part into "The vector tick (*Ixodes ricinus*) inoculates the infective sporozoites into the blood-stream of the host, where they invade erythrocytes, undergo repeated replication that differentiate into variety of morphology, then escape and invasion new erythrocytes, resulting in destruction of host red blood cells." in lines 68-70.

3. Line 66. What is meant by, "...value-added phase of apical compound parasites.."? It is not clear what the authors were saying.

Response: We have corrected the "Excellent *in vitro* culture systems can be used to study the value-added phase of apical compound parasites," into "Continuous *in vitro* culture systems can be used to study the proliferation stage of apicomplexan parasites," in lines 70-71.

4. Lines 68-70 could be deleted without loss.

Response: Line 68-70 has been deleted.

5. Line 73. Reference (2), referring to the seminal paper by Levy and Ristic, is not even included among the references.

Response: We have added this reference by Levy and Ristic in Reference (2).

6. Lines 80-81 could be deleted without loss.

Response: Line 80-81 has been deleted.

7. Lines 94-97 could be deleted without loss.

Response: Line 94-97 has been deleted.

8. Line 110. Title should read, "...continuous growth of *B. gibsoni*."

Response: Title has been re-written to "Culture initiation and continuous growth of *B. gibsoni*." In line 120.

9. Why was the experiment shown in Figure 1 not performed as a direct comparison of the two media compositions? It is not possible to directly compare the growth rates of parasites at different times in different media, perhaps with different dog sera. Given how this was done, it could be possible to perform a Z-test analysis on normalized rates of proliferation in the two media, and plotted as cumulative proportions. This would at least provide statistical support for the growth rates being equivalent (or not), although from an experimental point of view it is an inferior approach.

Response: *Babesia gibsoni* WH strain was initially isolated from the blood of infected dogs in Wuhan of China. The initial establishment of culture used different concentrations of serum, and only concentrations up to 40% can maintain the growth of the parasite. After a long period of adaption, the infection rate began to stabilize, then attempts to reduce the serum concentration to 20%, after a period of culture, the parasite began to adapt to this serum concentration. However, attempts to continue to reduce the serum concentration failed, we began to change the medium formula to achieve the purpose of reducing the serum concentration. The Figure 1B was a record of the parasitemia of this parasite *in vitro* culture for 380 days.

10. Figure 1. Could the authors explain why so many parasites have a highly vacuolated appearance?

Response: During *in vitro* culture of *B. gibsoni* in our study, the parasite with a highly vacuolated appearance always could be observed. In addition, morphological changes were also observed with a variety of cell morphologies appeared when parasitized erythrocytes were high. This behaviour is probably due to the absence of the normal immune system refer to the paper of Walter et al (2002). *B. bovis* and *B. bigemina* with a highly vacuolated appearance were also observed at high density using a perfusion bioreactor (Rojas-Martinez C, 2017,2018; Alvarez, 2020). It also could be observed in the continuous *in vitro* *B. bovis* culture with a medium free of animal origin components (Álvarez, 2021). And Photographs are often performed from cultures from high parasitemia to obtain sufficient data, which may be the main reason that so many parasites have a highly vacuolated appearance.

Giemsa-stained blood smear from *in vitro* cultured-derived *Babesia bovis*. (A) Conventional

culture system. (B) Bioreactor culture system. 100×.

Giemsa-stained blood smear from *in vitro* cultured-derived *Babesia bigemina*, (A) Conventional culture system. (B) Bioreactor culture system. 100×.

Large-scale *in vitro* proliferation of *B. bovis* in a hollow fiber perfusion bioreactor system (HFPBS).

(1) Walter, S., Mehlhorn, H., Zweygarth, E. et al. (2002). Electron microscopic investigations on stages of dog piroplasms cultured *in vitro*: Asian isolates of *Babesia gibsoni* and strains of *B. canis* from France and Hungary. *Parasitol Res* 88, 32–37. <https://doi.org/10.1007/s004360100495>

(2) Rojas-Martinez C, Rodriguez-Vivas RI, Figueroa-Millan JV, Acosta-Viana KY, Gutierrez-Ruiz EJ, Alvarez-Martinez JA. (2017). Putrescine: essential factor for *in vitro* proliferation of *Babesia bovis*. *Exp Parasitol*. 175:79–84. doi: 10.1016/j.exppara.2017.01.010

(3) Rojas-Martinez C, Rodriguez-Vivas RI, Figueroa-Millan JV, Acosta-Viana KY, Gutierrez-Ruiz EJ, Bautista-Garfias CR, et al. (2018). *Babesia bigemina*: advances in continuous *in vitro* culture using serum free medium, supplemented with insulin, transferrin, selenite and putrescine. *Parasitol Intl*. 67:294–301. doi: 10.1016/j.parintl.2017.11.003

(4) Alvarez JA, Rojas C and Figueroa JV (2020). An Overview of Current Knowledge on *in vitro* Babesia Cultivation for Production of Live Attenuated Vaccines for Bovine Babesiosis in Mexico. *Front. Vet. Sci*. 7:364. doi: 10.3389/fvets.2020.00364

(5) Álvarez Martínez JA, Figueroa Millán JV, Ueti MW, Rojas-Martínez C. (2021). Establishment of *Babesia bovis* *In Vitro* Culture Using Medium Free of Animal Products. *Pathogens*. 10(6):770. doi: 10.3390/pathogens10060770. PMID: 34205286; PMCID: PMC8235554.

11. Lines 131, 207. It would be helpful to state the cell cycling time for this parasite as measured *in vitro*.

Response: Thank you for the suggestion. We tried different ways to measure the cell

cycle period *in vitro*. However, we are not confident about the data. Because we could not synchronize *B. gibsoni* *in vitro* as *Plasmodium*, *B. divergens* and *B. bovis*. Using cryo-soft X-ray tomography (cryo-SXT), the visualization of a synchronized *B. divergens* asexual cycle for the first 24 h showed seven intraerythrocytic (IE) stages, including a single round trophozoite, paired pyriforms (two attached pear-shaped sister cells), double trophozoites (two round unattached cells), double paired pyriforms (two sets of paired sister cells), tetrads or Maltese Crosses (four attached sister cells), quadruple trophozoites (four round unattached cells), and multiple parasites (RBCs containing more than four parasites). However, 24 h later, the life cycle progresses and lose its synchronicity, transforming into asynchronous populations in highly parasitized RBCs (Cursino, 2016). Rossouw et al (Rossouw, 2015) hypothesize *B. divergens* propagation diagram based on the morphological observations, DNA measurements, temporal distribution and transcriptome expression dynamics. The significant increase in newly infected erythrocytes (ring formations) observed during the first 4–6 hours in culture is markedly quicker than previous reports where the *in vitro* life cycle of *B. divergens* parasites was claimed to last around 8 hours under the culture conditions.

The average doubling time of *B. bovis* was roughly 10 h for the purified merozoites that invaded RBCs, while 12.4 h was also observed, which could be due to the difference in culture conditions and the number of parasites that were visualized. Purified *B. bovis* merozoites started growing upon invasion and lost their synchronicity within one cycle (Hakimi, 2021).

Simplified model of the *B. divergens* asexual cycle: from the single trophozoite to the multiparasite stage. (Cursino, 2016)

Rossouw et al. hypothesize *B. divergens* propagation diagram based on the morphological observations, DNA measurements, temporal distribution and transcriptome expression dynamics. The significant increase in newly infected erythrocytes (ring formations) observed during the first 4–6 hours in culture is

markedly quicker than previous reports where the *in vitro* life cycle of *B. divergens* parasites was claimed to last around 8 hours under the culture conditions.

Model of the intra-erythrocytic developmental cycle of *B. divergens* parasites. (Rossouw, 2015)

Time-lapse imaging of *B. bovis*. (Hakimi, 2021)

Cursino-Santos, J. R., Singh, M., Pham, P., Rodriguez, M., & Lobo, C. A. (2016). *Babesia divergens* builds a complex population structure composed of specific ratios of infected cells to ensure a prompt response to changing environmental conditions. *Cellular microbiology*, 18(6), 859–874. <https://doi.org/10.1111/cmi.12555>

Rossouw, I., Maritz-Olivier, C., Niemand, J., van Biljon, R., Smit, A., Olivier, N. A., & Birkholtz, L. M. (2015). Morphological and Molecular Descriptors of the Developmental Cycle of *Babesia divergens* Parasites in Human Erythrocytes. *PLoS neglected tropical diseases*, 9(5), e0003711. <https://doi.org/10.1371/journal.pntd.0003711>

Hakimi, H., Asada, M., Ishizaki, T., & Kawazu, S. (2021). Isolation of viable *Babesia bovis* merozoites to study parasite invasion. *Scientific reports*, 11(1), 16959. <https://doi.org/10.1038/s41598-021-96365-w>

12. Lines 138-139 (and throughout). The data should be stated with calculated s.d. or s.e. values. When it is stated that, "..growth rates of *B. gibsoni* in the three group had no difference.." no measure of reproducibility or statistical support was given for this statement.

Response: Thanks for your suggestion, we have added statistical analysis in the figures and the newly submitted manuscript.

13. Figure 2. Labeling on this figure is much too small to read. Also, it is incorrect to refer to the five subcultures as "..five consecutive generations..". Over 15 days these parasites likely went through 30-50 generations.

Response: We have modified figure labeling in Figure 2, and we have changed "five consecutive generations" to "five subcultures".

Figure 2. Gradual lowering of the serum concentration and Albumax I in the culture medium for its elimination from the in vitro culture of *B. gibsoni*.

14. Figure 3B. Figure labeling is much too small to read.

Response: We have modified the figure labeling in Figure 3B.

15. Lines 187-190. The statement about octoploids arising from daughter merozoites not leaving the cell may be true, but some form of actual experimental evidence is required to support that statement. The alternative, that multiple merozoites can invade the same erythrocyte, was not ruled out.

Response: Thanks for your suggestion. We agree with your idea, and the description of "The statement about octoploids arising from daughter merozoites not leaving the cell" have deleted.

16. Figure 5. What is the source of the mCherry fluorescence? Clearly, these are genetically modified parasites, which is incompatible with the description of parasites in the Methods section. The nature of the modifications and the cultivation history of

these parasites must be identified, particularly as these parasites have previously undergone significant in vitro culture adaptation. Also, there is a significant pixel-shift problem with Figure 5B where light and fluorescence images do not align properly. This must be corrected.

Response: Considering the Reviewer's suggestion, the sections for the generation of Delta Bgef1a-b transgenic parasites by electroporation and the section have been provided in the "Materials and Methods" (lines 404-420) and "Result" (lines 246-255). During the acquisition of fluorescence images, the parasites are alive and able to move in red blood cells, so there may be a significant pixel-shift problem using living parasites' fluorescence. We have done our best to correct this problem.

Lines 246-255: The plasmid (pBS-ef1a-B) expressing mCherry and PAC was constructed, using 750-bp 5'UTR and 3'UTR of ef-1 α B as the promoter and terminator, respectively (Figure 5A). At 12 days post-selection by 4 nM Puromycin, mCherry-expressing parasites appeared in cultures transfected with plasmids (Figure 5B). PCR1 (F1R1), PCR2 (F1R2), and PCR3 (F2R1) primer pairs could successfully amplify 3346, 2329, and 2322 bp fragments in the Δ Bgef1a-B strain, respectively, but not in the wild-type (WT) of *B. gibsoni* (Figure 5C). Western blot analyses indicated the expression of protein mCherry-PAC in Bgef1a-b knockout strains (Figure 5D).

Lines 404-420: Plasmid constructs and Transfection of parasites. The schematic diagram of the plasmid (pBS-ef1a-B) used in this study is shown in Figure 5B. The reporter gene (mCherry) and drug selection gene (Puromycin) cassettes were driven with Bgef1 α 5' UTR (IG-B). Bgef1 α 5' UTR (IG-B) and Bgef1 α 3' UTR were used as recombination sites cloned into the upstream and downstream of the mCherry-PAC genes. All the PCR primer pairs used for plasmid construction are listed in Table 2. The constructed plasmid was purified using Qiagen® Plasmid Maxi Kit (Qiagen, Hilden, Germany) according to the manufacturer's instructions, and was confirmed by sequencing before transfection. Transfection of parasites as previously described (26). In brief, *B. gibsoni*-infected red blood cells (iRBCs) were centrifuged at low speed and then the culture supernatant is removed. linearized pBS-ef1a-B plasmid (20 μ g) and iRBCs were mixed and transfected using Lonza buffer SF and program FA113 of Amaxa 4D Nucleofector™ device (Lonza, Cologne, Germany). The transfected mixtures were immediately transferred into a preheated culture containing 10% fresh RBCs. 4 nM Puromycin (4nM) was used to select mCherry-expressing transgenic parasites. After 3 weeks of drug selection, the parasite population was cloned in a 96-well culture plate using limiting dilution as previously described.

Figure 5. Establishment of the *Bgef1a-b* knockout strains with stable expression of mCherry-PAC.

17. Line 224. Reference 15 does not refer to continuous culture. This was not accomplished with *Babesia* spp. until Levy and Ristic (1980) achieved this for *B. bovis*.

Response: We have changed this reference according to the reviewer's suggestion.

18. Line 233. "microaerobic.." should be "microaerophilous..", and the correct reference is again Levy and Ristic (1980).

Response: We have changed this word and reference according to the reviewer's suggestion.

19. Line 291. It should be stated when the parasites were isolated in Wuhan, and any prior cultivation history. This sounds as though the authors were working with a primary isolate, but this is clearly not the case. Figure 5 dispels that idea.

Response: As suggested by the reviewer, we have added more information about this parasite. "*B. gibsoni* WH strain was originally isolated from a naturally infected dog in Wuhan of China in 2017. The strain underwent a 280-day domestication process before stable in vitro culture, during which the growth of the parasite was unstable with the highest parasitemia changed greatly from 3-8%, and the subculture cycle was 5-8 days." in lines 361-365.

20. Table 1 could be eliminated, as the information is provided in the text. Alternatively, refer to Table 1 and eliminate most of the associated verbiage from the text.

Response: Thanks for your excellent suggestion. Table 1 was retained and most of the associated verbiage was eliminated from the text.

21. Line 334. "emersion" should be "immersion".

Response: We have changed this word according to the reviewer's suggestion.

Reviewer #2 (Comments for the Author):

The manuscript "Establishment of continuous in vitro culture of *Babesia gibsoni* by using VP-SFM medium with low concentration serum" by Li et al. describes the methodology for a long-term in vitro culture of a dog *Babesia* species. The article is well written, but some statistical and technical details are missing particularly for the subcloning.

Response: Thanks very much for reviewing this manuscript. I really appreciate all your comments and suggestions. Please find my itemized responses in below and my revisions in the re-submitted files.

Major points:

Figure 2B and 2C: The authors must add the statistical analysis.

Response: Figure 2 has been revised to better present the statistical analysis.

Figure 5. Establishment of the *Bgef1a-b* knockout strains with stable expression of mCherry-PAC.

In the result par on the derivation of subclone of Delat *Bgef1a-b* isolate, the authors must have full genomic data to demonstrate the success of the subclone isolation.

Response: Thanks very much for the reviewer's comments, the double-cope gene *ef1a* including *ef1a-A* and *ef1a-B*, which started by a bidirectional promoter (*ef1a-IG*). It cannot be verified with genomic data in the way that conventional methods, such as the ORF of *ef1a-B* were amplified by PCR because the sequences of *ef1a-A* and *ef1a-B* were identical and reversed. We have added "The plasmid (pBS-*ef1a-B*) expressing mCherry and PAC was constructed, using 750-bp 5'UTR and 3'UTR of *ef-1α B* as the promoter and terminator, respectively (Figure 5A). At 12 days post-selection by 4 nM Puromycin, mCherry-expressing parasites appeared in cultures transfected with plasmids (Figure 5B). PCR1 (F1R1), PCR2 (F1R2), and PCR3 (F2R1) primer pairs could successfully amplify 3346, 2329, and 2322 bp fragments in the $\Delta Bgef1a-B$ strain, respectively, but not in the wild-type (WT) of *B. gibsoni* (Figure

5C). Western blot analyses indicated the expression of protein mCherry-PAC in Bgef1a-b knockout strains (Figure 5D).” in lines 246-255.

Figure 5B: The authors need to add flow cytometry data to show the purity of the fluorescent subclone strains.

Response: As suggested by the reviewer, we have added flow cytometry data in line and Figure 6B. Red fluorescence was detected in almost all parasites using flow cytometry, which indicated that parasites obtained using VP-SFM AD (2.5%) medium were monoclonal after 18 days of culture without drug screening (Figure 6B). As shown in lines 261-264.

Lines 261-264: In addition, red fluorescence was detected in almost all parasites using flow cytometry, which indicated that parasites obtained using VP-SFM AD (2.5%) medium were monoclonal after 18 days of culture without drug screening (Figure 6B).

Figure 6. Limiting dilution cloning of parasites using VP-SFMAD (2.5%).

Materials and Methods: The sections for the generation of Delta Bgef1a-b transgenic parasites by electroporation and the section for the fluorescence microscopy are missing.

Response: As suggested by the reviewer, we have added the sections for the generation of Delta Bgef1a-b transgenic parasites by electroporation and the section for the fluorescence microscopy in “Materials and Methods”. As showed in lines 404-420.

Lines 404-420: Plasmid constructs and Transfection of parasites. The schematic diagram of the plasmid (pBS-ef1a-B) used in this study is shown in Figure 5B. The reporter gene (mCherry) and drug selection gene (Puromycin) cassettes were driven with Bgef1α 5’ UTR (IG-B). Bgef1α 5’ UTR (IG-B) and Bgef1α 3’ UTR were used as recombination sites cloned into the upstream and downstream of the mCherry-PAC genes. All the PCR primer pairs used for plasmid construction are listed in Table 2. The constructed plasmid was purified using Qiagen® Plasmid Maxi Kit (Qiagen,

Hilden, Germany) according to the manufacturer's instructions, and was confirmed by sequencing before transfection. Transfection of parasites as previously described (26). In brief, *B. gibsoni*-infected red blood cells (iRBCs) were centrifuged at low speed and then the culture supernatant is removed. linearized pBS-ef1a-B plasmid (20µg) and iRBCs were mixed and transfected using Lonza buffer SF and program FA113 of Amaxa 4D Nucleofector™ device (Lonza, Cologne, Germany). The transfected mixtures were immediately transferred into a preheated culture containing 10% fresh RBCs. 4 nM Puromycin (4nM) was used to select mCherry-expressing transgenic parasites. After 3 weeks of drug selection, the parasite population was cloned in a 96-well culture plate using limiting dilution as previously described.

Minor points:

Line 42: Define PPE, it should be for parasitized erythrocytes.

Response: We have changed “PPE” to “parasitized erythrocytes”.

Figure 1B: The authors should highlight in the figure the days with 40% and 20% DS and in the legend specify when fresh RBC are added to decrease the parasitemia.

Response: As suggested by the reviewer, we have changed Figure 1. They are data from in vitro culture that lasted 380 days, and fresh RBCs are added to decrease the parasitemia when the parasite grows for 3 or 4 days.

Figure 2B: The authors must add the number of biological replicates in the legend.

Response: We have added “Data represent mean values measured in triplicates.” in the legend.

Line 308: Any references to cite showing that penicillin has no effect on Babesia parasite development?

Response: We did not know if penicillin has effect on *Babesia* parasite development. While Penicillin has potent combined antimicrobial effects against gram-positive and gram-negative bacteria and can be used to prevent bacterial contamination of cell cultures. The use of Penicillin (100 U/mL) was a reference to previous culture protocols in *Toxoplasma* and *Babesia*.

Christiansen, C., Maus, D., Hoppenz, E., Murillo-León, M., Hoffmann, T., Scholz, J., Melerowicz, F., Steinfeldt, T., Seeber, F., & Blume, M. (2022). In vitro maturation of *Toxoplasma gondii* bradyzoites in human myotubes and their metabolomic characterization. *Nature communications*, 13(1), 1168. <https://doi.org/10.1038/s41467-022-28730-w>

Wang, S., Li, D., Chen, F., Jiang, W., Luo, W., Zhu, G., Zhao, J., & He, L. (2022). Establishment of a Transient and Stable Transfection System for *Babesia duncani* Using a Homologous Recombination Strategy. *Frontiers in cellular and infection microbiology*, 12, 844498. <https://doi.org/10.3389/fcimb.2022.844498>

March 9, 2023

Prof. Junlong Zhao
Huazhong Agricultural University
College of Veterinary Medicine
Shizishan street 1
Wuhan, Hubei 430070
China

Re: Spectrum00258-23R1 (Establishment of continuous *in vitro* culture of *Babesia gibsoni* by using VP-SFM medium with low concentration serum)

Dear Prof. Junlong Zhao:

The revised manuscript has been reviewed and the improvements are noted. However, several of reviewer #1's concerns remain and these include the need for statistical inputs, clarity in statements, more details for plasmid construction and improved image overlays. The authors are requested to further improve their manuscript before a decision on its suitability for publication can be considered.

Link Not Available

Sincerely,

Kevin Tan

Journals Department
Reviewer comments:

Reviewer #1 (Comments for the Author):

This revised manuscript is now improved over the original. The authors are thanked for addressing most of the concerns of the reviewers, although some remain. Despite these improvements there remain points in need of improvement before this manuscript could be ready for publication. Most of these points, listed below, could be addressed at the keyboard. A few require

more work.

[1] line 63. Unlike mosquitoes and malaria, ticks transmitting Babesia do not "inoculate sporozoites into the bloodstream..". In tick feeding, the mouthparts create a damaged dermal wound into which blood and lymph pool, and into which sporozoites are released. Sporozoites must find their way to the circulation from that pool. Please rewrite this statement for accuracy.

[2] line 120. I initially misunderstood the nature of the results shown in Figure 1. The authors' description is now clearer. Regardless, statistical support should be provided comparing the relative growth rates of the parasites in the two different media, based upon recorded growth records and dilutions involved in subcultures. This would allow a more rigorous statement of comparative growth during establishment and between media.

[3] line 138 (new ms). What is meant by, "..instability in contamination rates.."? This is not clear as written, nor in context. Contamination would refer to the growth of something unintended in the culture, for example bacterial or fungal growth.

[4] lines 146-147. The authors still refer to "..five consecutive generations..", despite stating they had corrected that to "five subcultures".

[5] lines 205, 358. At least skeletal detail should be given about the construction of plasmid pBS-ef1a-B, and the accession number for EF1a-B from the reference genome.

[6] original comment 11. It would still be helpful to state an approximate cell cycling time for this parasite, even if not completely accurate, to help the reader assess and anticipate growth characteristics.

[7] original comment 13. The labeling on all figures remains too small and needs to be redone.

[8] original comment 16. The authors have not addressed this concern satisfactorily. There remains a significant pixel-shift problem wherein the DIC and various fluorescent images do not align properly. This is not due to parasite movement, which would manifest as blurred images, but rather to non-identical redirection of light by the fluorescence filter cubes relative to the plane of the image.

Reviewer #2 (Comments for the Author):

No further comments.

Staff Comments:

Preparing Revision Guidelines

Please return the manuscript within 60 days; if you cannot complete the modification within this time period, please contact me. If you do not wish to modify the manuscript and prefer to submit it to another journal, please notify me of your decision immediately so that the manuscript may be formally withdrawn from consideration by Microbiology Spectrum.

Reviewer #1:

This revised manuscript is now improved over the original. The authors are thanked for addressing most of the concerns of the reviewers, although some remain. Despite these improvements there remain points in need of improvement before this manuscript could be ready for publication. Most of these points, listed below, could be addressed at the keyboard. A few require more work.

Response: Thank you very much for your constructive comments and suggestions which would help us in depth to improve the quality of the paper.

[1] line 63. Unlike mosquitoes and malaria, ticks transmitting Babesia do not "inoculate sporozoites into the bloodstream..". In tick feeding, the mouthparts create a damaged dermal wound into which blood and lymph pool, and into which sporozoites are released. Sporozoites must find their way to the circulation from that pool. Please rewrite this statement for accuracy.

Response: According to the reviewer's suggestion We have re-written this part to in lines 56-60: "During tick feeding, sporozoites are transmitted through saliva secretion to the vertebrate, subsequently into the bloodstream and invade erythrocytes, undergo repeated replication that differentiates into variety of morphology, then escape and invasion of new erythrocytes, destroying host red blood cells."

[2] line 120. I initially misunderstood the nature of the results shown in Figure 1. The authors' description is now clearer. Regardless, statistical support should be provided comparing the relative growth rates of the parasites in the two different media, based upon recorded growth records and dilutions involved in subcultures. This would allow a more rigorous statement of comparative growth during establishment and between media.

Response: Thank you for the suggestion. Statistical analyses were performed, and the relative growth rates of the parasites in the two different mediums were compared (Figure 1). The growth rate of parasites were 2% (± 0.6) and 2.4% (± 0.6) in RPMI-1640D(40%) and RPMI-1640D(20%) medium, respectively. The average growth rate of every day in each subculture is obtained by the PPE of last day subtracting the PPE after been diluted involved in the previous subculture, then dividing the number of days cultivated in each subculture. In addition, the doubling time = 24h / (the growth rate/ the diluted PPE). The average growth rate using RPMI-1640D (20%) is fast than RPMI-1640D (40%) with significant differences ($p = 0.0175$), which suggests that the parasite has adapted to RPMI-1640D (20%). Those descriptions have been added in lines 109-112 and 124-128.

(A)

(B)

(C)

[3] line 138 (new ms). What is meant by, "..instability in contamination rates.."? This is not clear as written, nor in context. Contamination would refer to the growth of something unintended in the culture, for example bacterial or fungal growth.

Response: Thank you for the suggestion. The sentences were updated to "However, when using VP-SFMAD (2.5%) medium for *B. gibsoni* *in vitro* culture, instability in multiplication rates were observed." in lines 149-151.

[4] lines 146-147. The authors still refer to "..five consecutive generations..", despite stating they had corrected that to "five subcultures".

Response: Sincerely apologize for the mistake. We have double checked the entire manuscript, all "consecutive generations" were corrected to "five subcultures". in lines 160 and 165.

[5] lines 205, 358. At least skeletal detail should be given about the construction of plasmid pBS-ef1a-B, and the accession number for EF1a-B from the reference genome.

Response: Skeletal detail have been added in lines 223-225, with "Bgef1a-B 5'UTR and Bgef1a-B 3'UTR were used as recombination sites cloned into the upstream and downstream of mCherry-PAC gene to construct plasmid pBS-ef1a-B".

"The schematic diagram of the plasmid (pBS-ef1a-B) used in this study is shown in Figure 5A, and sequence of plasmid pBS-ef1a-B has been uploaded in National Center for Biotechnology Information (NCBI) database with the accession number OQ630424. The Bgef1a-B gene (BgWH_03g00751) located on chromosome 3 of *B. gibsoni*. with the accession number OQ630425 in NCBI." has been added in lines 375-380.

[6] original comment 11. It would still be helpful to state an approximate cell cycling time for this parasite, even if not completely accurate, to help the reader assess and anticipate growth characteristics.

Response: Approximate. cell cycling time was calculated and the data was added into the manuscript in lines 112-114: "According to the growth rate, the doubling time of *B. gibsoni* was about 6.4 – 13.7 h in the asexual stage".

[7] original comment 13. The labeling on all figures remains too small and needs to be redone.

Response: We have updated all the figures according to your suggestions. The size of labeling are all increased.

[8] original comment 16. The authors have not addressed this concern satisfactorily. There remains a significant pixel-shift problem wherein the DIC and various fluorescent images do not align properly. This is not due to parasite movement, which would manifest as blurred images, but rather to non-identical redirection of light by the fluorescence filter cubes relative to the plane of the image.

Response: We believe your comments are reasonable. We tried our best, repeated the experiment and took new images after received your comment. Here is the procedure how we prepare the figures, hope it can help us. Fluorescence images were acquired using an Olympus BX53 microscope, and images were processed using Zeiss ZEN Microscopy Software. First, the original pictures of the three channels were showed with pseudo coloring and merged, and then the region of interest was drawn with box.

Previous images were acquired using Olympus all-electric BX63 microscope, which recently is undergoing an recondition.

April 4, 2023

Prof. Junlong Zhao
Huazhong Agricultural University
College of Veterinary Medicine
Shizishan street 1
Wuhan, Hubei 430070
China

Re: Spectrum00258-23R2 (Establishment of continuous *in vitro* culture of *Babesia gibsoni* by using VP-SFM medium with low concentration serum)

Dear Prof. Junlong Zhao:

The authors have addressed the reviewer's comments satisfactorily after 2 rounds of revision.

Your manuscript has been accepted, and I am forwarding it to the ASM Journals Department for publication. You will be notified when your proofs are ready to be viewed.

Sincerely,

Kevin Tan
Editor, Microbiology Spectrum
